# Challenges Facing Undergraduate Medical Education in Ambulatory Care Clinics at Tertiary Care Hospitals

**DOI:** 10.3390/healthcare10030496

**Published:** 2022-03-08

**Authors:** Youssef B. Almushait, Mohamad S. Alabdaljabar, Khalid Alkhani, Hesham M. Abdalla, Raid Alhayaza, Mohamad-Hani Temsah, Fahad Alsohaibani

**Affiliations:** 1College of Medicine, Alfaisal University, Riyadh 11211, Saudi Arabia; yalmushait7@gmail.com (Y.B.A.); msaljabbar@gmail.com (M.S.A.); heshamufc@gmail.com (H.M.A.); raidalhayaza93@gmail.com (R.A.); 2College of Medicine, King Saud University, Riyadh 11211, Saudi Arabia; koalkhani@gmail.com (K.A.); mtemsah@ksu.edu.sa (M.-H.T.); 3Department of Medicine, King Faisal Specialist Hospital & Research Center, Riyadh 11211, Saudi Arabia

**Keywords:** outpatient, medical education, barriers, Saudi Arabia, medical students

## Abstract

**Background**: Medical education has been rapidly growing and transforming due to the enormous evolution of medicine. There have been many proficient ways to learn in medicine, but academic lectures, attending wards, and ambulatory care clinics (ACC) remain the three main ways of gaining clinical knowledge and experience for medical students. Over the past decade, there has been a dramatic shift in care by focusing on ambulatory care rather than inpatient settings, which provides a golden opportunity to reinforce medical education. **Purpose**: Most of the published studies that have focused on the teaching barriers in ACC were descriptive rather than analytic studies. Herein, we aim to detect and determine the barriers to teaching in ACC settings using qualitative analysis. **Methods**: This is a cross-sectional, observational study, involving medical students in their clerkship years (i.e., fourth and fifth) from two different medical colleges in Riyadh, Saudi Arabia. Faculty who are involved in undergraduate medical education in both colleges were also included. **Main Results**: A total of 387 medical students studying at the two universities were enrolled in the study. Most of the participants preferred attending outpatient clinics with consultants (44.2%) and the majority preferred attending internal medicine (IM) and IM subspecialties clinics (40.4%). Regarding the challenges, students believe the top three barriers are related to: faculty (39%), environment (34.8%), and patients (14.8%). Faculty on the other hand see that the top three barriers are related to environment (55.6%), patients (24.4%), and faculty (20%). **Conclusion**: Undergraduate medical education in outpatient settings has many challenges. In our study, the most significant challenges were COVID-19-related restrictions, patient refusal, and insufficient time for teaching. Future studies are needed to investigate these barriers and explore potential solutions that can decrease their burden on undergraduate medical education.

## 1. Introduction

Understanding the best means to enhance medical education has been a primary goal for many educators and researchers. During the past decade, many changes in the healthcare system have been developed. One of the most significant changes was the dramatic shift of care towards ambulatory rather than the traditional inpatient care settings [1]. Since inpatient cases are more likely to be critically ill or under subspecialty conditions, they are less representative of common case presentation in clinical practice. This emphasizes the importance of clinical education in ambulatory care settings in addition to inpatient settings. Unfortunately, many programs have been described as flawed in outpatient training and required major changes [2].

The ways that stand the most are academic lectures, attending wards, and ambulatory care clinics (ACC). All of these ways positively impact the educational experience of medical students throughout their journey. ACC have been a crucial method in learning, for several reasons [3]. One of the most important reasons is that it offers an effective environment for medical learning where the chance is given for students to encounter a broad spectrum of pathologies in various evolutionary phases [4]. In addition, ACC play a significant role in the establishment and improvement of the student’s communication and clinical skills, including history taking, clinical reasoning, and performing physical examination [1,3].

Approximately 80% of clinical education takes place in the inpatient settings, yet about 80–90% of practiced medicine is in the outpatient settings [5]. The current practice allows untrained medical students to begin their clinical training in inpatient settings by being exposed to complicated cases. This has led many professional societies to call for ambulatory training reforms, including the American College of Physicians (ACP) and the Society of General Internal Medicine (SGIM) [6]. However, concerns have been raised about the readiness and willingness of undergraduate programs to implement such changes [7,8].

Recent studies have outlined several barriers in ACC that can impact the learning process for medical students and for the academic staff in a negative way. Franco et al. divided the barriers into four categories: student-related, staff-related, patient-related, and environment-related [3]. There is a clear gap in the medical literature that explores teaching barriers in ACC for medical students in their clerkship years.

In this study, we aim to evaluate the challenges that medical students could face in their clerkship years at ACC in two different tertiary care hospitals from both, students’ and physicians’ perspectives. We also plan to contrast these challenges with what has been published in the literature and to provide future directions on how to improve the ACC teaching experience.

## 2. Methodology

### 2.1. Study Design and Population

This is an observational cross-sectional study that was conducted at two major medical colleges in Riyadh, Saudi Arabia namely: Alfaisal University (AU) with King Faisal Specialist Hospital & Research Center (KFSH&RC) as the clinical training center, and King Saud University (KSU) with King Khaled University Hospital (KKUH) as the clinical training center. This study was conducted between 1 March 2020 and 30 April 2021. The study involved two types of participants: medical students and faculty members from both colleges. The inclusion criteria for the medical students included 4th and 5th year of undergraduate program attending the college of medicine at either AU or KSU who completed a minimum of 12 weeks of clinical clerkship. The inclusion criteria for the faculty members included faculty members of AU or KSU who are actively involved in ACC education.

### 2.2. Sample Size

A convenience sampling technique of 405 and 375 participants for the medical student sample was determined from AU and KSU medical students, respectively. As for the faculty, a convenience sampling technique of 297 and 180 participants was recruited from AU and KSU, respectively. The sample size was calculated by Asher Sample Size Calculator.

### 2.3. Data Collection/Instrument

The email list of all eligible medical students and faculty members from both medical schools was obtained from administration offices at both sites. An email template for the medical student questionnaire and another email template for faculty members were prepared. Since the current literature lacks a validated tool that could be used specifically to assess challenges that students might face in ACC, we used the literature to build this tool based on potential barriers. Questionnaire validation was subject to a panel of experts for content and face validity. An email was sent to five out of the six experts who evaluated the instrument and had agreed on the value and the usefulness of the used instrument. The experts forwarded their feedback via emails and it is beyond our ability to retrieve and perform Kendall’s W test at this stage. Unfortunately, because the survey was being amended as the email feedback was received from the panel, emails were not kept in record.

The questionnaire for medical students included three main domains: demographic information, perception of general barriers to ACC education, and perception of specific barriers. Emails were sent out to eligible medical students and complete responses were included in the study. Questionnaires emailed to faculty members included three main domains: socio-demographic information, perception of ACC education, and perception of barriers to ACC teaching. Emails were sent out to eligible faculty members and complete responses were included in the study. Both student and faculty surveys can be found in the Appendix A.

### 2.4. Data Analysis

Data were analyzed using SPSS statistical software. Descriptive statistics (mean, standard deviation, frequencies, and percentages) were used to describe the quantitative and categorical variables. The Kolmogorov–Smirnov test of statistical normality and the Histograms were used to assess the statistical normality of the measured metric variables, and Levene’s test was applied to assess the statistical equality of variance assumption. The tested metric variables were found approximately normally distributed with no violation of homogeneity of variance statistical assumption.

Pearson’s chi-square test was used to assess the association between the categorical variables. Student’s *t*-test was used to compare mean values. A *p*-value ≤ 0.05 and 95% confidence intervals were used to report the statistical significance and precision of the results (Figure 1).

### 2.5. Ethical Considerations

The study received approval from the office of research affairs and ethics committees of KFSH&RC (Project Number: 2211197) and KSU (Project Number: E-22-6592) and was conducted according to the hospitals’ bylaws. Participation in this study is voluntary, and no personal identifiers were collected. Consent to participate is incorporated in the survey cover page, and data were stored under password protection.

## 3. Results

In this section, we will discuss the results of both student and faculty surveys on the learning process at ACC. In each of the two sections (Section 3.1 and Section 3.2), we will go over the characteristics, general perception, in addition to environment-, faculty-, patient-, and student-related barriers. In the student survey, we will also evaluate how receptive students are to virtual clinics (VC), and we will discuss their preferred clinic (based on specialty).

### 3.1. Students Survey Analysis

#### 3.1.1. Demographic Factors

A total of 405 eligible students attending AU and 375 eligible students attending KSU were invited to participate in the survey. The response rate for each medical school was 37.8% and 62.4%, respectively. The total number of medical students who agreed to participate and completed the survey was 387 (153 and 234 from AU and KSU, respectively). Female participants were seen in 45.7% of the whole sample with 62.1% and 35% were female students from AU and KSU respectively. Fourth-year medical students participated in 49.1% (54.2% and 45.7% from AU and KSU, respectively) (Table 1).

#### 3.1.2. Student’s General Perception

Most students preferred attending clinics with consultants (44.2%) followed by senior residents (25.3%) and fellows (15.5%). Thirty-nine percent considered faculty related-barriers to be the main source of challenges. While institutional-related barriers were perceived as a challenge by 34.8% of the sample. Participants attending KSU perceived significantly (*p* = 0.022) less faculty-related barriers to outpatient ACC learning compared to participants from AU. Participants from KSU were more likely to perceive COVID-19-related factors as a barrier than AU participants (*p* = 0.018). Perceived patient-related barriers, medical student-related barriers, and other factors did not show significant differences between the two groups.

#### 3.1.3. Virtual Clinics and Preferred Clinics

When asked to rate how much they support involving medical students in VCs using a 1–5 Likert-like scale, with 5 denoting a great extent of support, the results showed that the students’ overall mean support level was measured with 3.1/5 points, SD = 1.44 points, signifying that these students had between a moderate-to-great extent of agreement with learning engagement in VCs. In an independent sample, the *t*-test showed that the AU medical students expressed greater support to the virtual clinical sessions (mean = 3.28) as compared to the KSU medical students (mean = 2.94, *p* = 0.026). Attending internal medicine (IM) and IM subspecialties clinics were preferred by 40.4% followed by Ob-Gyn (20.9%), then pediatrics (20.1%) (Table 2).

#### 3.1.4. Environmental Barriers

Looking closer at environmental factors (Table 3), “restrictions due to COVID19” was perceived as a barrier by more than half (52.5%) of the participants. Lack of structured teaching objectives was the second most perceived barrier to ACC learning according to 24.3% of the participants. Nineteen percent of the sample also perceived “inadequate distribution of students among ambulatory clinics” as a barrier to ACC learning, with only (4.1%) considering “inappropriate or small clinic rooms” as a challenge.

#### 3.1.5. Faculty-Related Barriers

The most perceived barrier by the sample in the faculty-related section was the “inadequate supervision and teaching by faculty” (48.6%). This is followed by “lack of time for teaching by doctors due to intense patient agenda” and “inappropriate or absent feedback”, which was perceived as a barrier by (33.9%) and (12.1%) of the participants, respectively. The variability between the two groups in the perception of faculty-related barriers did not yield a statistical significance, as (49.1%) of KSU participants and (47.7%) of AU participants perceived “inadequate supervision and teaching by faculty” as a barrier to ACC learning. The ranking of the rest of the perceived faculty-related factors was similar in participants from both universities.

#### 3.1.6. Patient-Related Factors

Around 49% of the entire sample perceived “patient refusal to be seen by medical students” as a patient-related factor limiting their ACC learning experience. While “no follow-up/continuity of care for cases attended” and “lack of suitable patients for teaching” were considered a barrier by (25.3%) and (25.1%) of the sample, respectively. In a comparison between the two universities, (51.3%) of KSU participants and (47.1%) of AU participants perceived patient refusal as a barrier. Participants from both universities had similar perceptions in terms of patient-related factors.

#### 3.1.7. Student-Related Factors

The most perceived student-related barrier was “the increasing number of students attending the ambulatory clinic” (39%) and (30.5%) of the sample perceived “not enough time to attend clinics” as a barrier to their ACC learning experience. Some students believed “lack of student commitment” (18.6%) and “no additional information gained compared to inpatient setting” (11.9%) to be a barrier to their ACC learning.

The four different groups of barriers significantly differed between students in the two universities, as relates to their unique characteristics (Table 4).

### 3.2. Faculty Survey Analysis

#### 3.2.1. Demographic Factors

Ninety faculty members from both KSU and AU agreed to participate and completed the questionnaire (68.9% of the participants were from AU, while the rest were from KSU). Males were 83.3% of the participants. Most of the responses were from associate professors (35.6%), followed by assistant professors (33.3%). Professors shaped 26.7% of the sample, with the least participants being “lecturers” (4.4%). The most common specialty of the participants was internal medicine and its subspecialties (43.3%), followed by pediatrics (23.3%), then neurosciences (6.7%) (Table 5).

#### 3.2.2. Faculty’s General Perception

When participants were asked how comfortable they were with medical students attending clinics with them, 35.6% of participants answered “uncomfortable” and 25.6% answered, “somewhat comfortable”. The rest of the sample (38.9%) ranged from “moderately comfortable” to “very comfortable”. The overall comfort rating was rated with 2.42 points out of 5 points, which highlights a moderate comfort perceived by participants. Participants from KSU and AU did not differ in their comfort toward medical students attending clinics and 93.3% from both universities believe that outpatient experience is important for medical students and should be integrated into their curriculum.

The environmental- and institutional-related barriers were believed to be the main barrier to proper and efficient medical student education in outpatient settings (55.6%) (Figure 2). While patient-related variables were considered a barrier by 24.4% of the participants, followed by faculty-related variables, which were considered a barrier by 20% of the sample. The two groups did not differ in the general perception of barriers toward medical student education in outpatient settings. Around one-third of the faculty (36.7%) support the involvement of medical students in virtual clinics to a very great extent. The overall support rating was 3.44 out of 5, indicating that the sample supported medical students attending virtual clinics to a great extent. An independent sample *t*-test suggested that the medical teachers at KSU may have significantly lower levels of support for involving students in virtual clinics (mean = 2.96) compared to the teachers in AU (mean = 3.66), *p* = 0.045 (Table 6).

#### 3.2.3. Environmental Factors

The most common environmental factor (Table 7) perceived as a barrier by the participants was restrictions due to COVID-19 (26.7%). In addition, 21.1% of the sample considered inadequate structuring and distribution of students in ACC as a barrier, and another 21.1% perceived “inappropriate or small clinics rooms” as a barrier toward medical teaching. Comparing the environmental factors between the two sites did not show statistical significance.

#### 3.2.4. Faculty Related Factors

Seventy percent of participants considered “insufficient time for teaching due to intense patient agenda” to be a barrier to medical teaching in outpatient settings. While 20% believed that medical teaching not being part of their key performance indicators (KPI) as a barrier toward medical teaching.

#### 3.2.5. Patient Related Factors

Patient refusal was perceived as a barrier toward medical teaching in outpatient settings by 41.1% of participants, while 24.4% of participants perceived “fear of patient dissatisfaction” as a barrier toward medical teaching. When the participants from the two sites were compared, no statistical significance was found.

#### 3.2.6. Student Related Factors

Half (51.1%) of participants believed that the “lack of student commitment and interest in learning” is a barrier toward medical teaching in outpatient settings. While another 48.9% believed that “the increasing number of students” is a barrier toward medical teaching in outpatient settings. When comparing between the two universities, perceived student-related barriers did not significantly differ between the two groups.

The differences in the perceived barriers between the faculty members of both universities are summarized in (Table 8).

## 4. Discussion

Several medical schools around the world have expanded and further developed the education process through ACC [9]. In the US, medical students spend at least a third of their total clinical experience in ACC [10]. While all methods of medical education are needed and each contributes differently, the teaching methods used in ACC are different from those commonly exercised in medical wards [11,12]. In most wards, the teaching rounds of inpatients are usually scheduled and are limited to a specific duration that is usually focused on a specific agenda [13]. Medical students are usually asked to examine patients who have already been diagnosed and have a treatment plan established. However, encounters in ACC can be related to either acute or chronic issues, giving students a broader perspective and exposing them to basic, yet essential clinical scenarios. This can aid in enhancing the learning experience for students and expose them to a wide variety of medical conditions [14,15].

The inadequate time of outpatient teaching and the reliance on unsuitable methods of instruction can be extremely hampering [16]. Adams et al. classified the ambulatory teaching costs that have been cited in the literature into seven categories: (1) increased office time for teaching purposes, (2) reduced productivity of the teaching site or teaching physician, (3) elevated operation costs, (4) trainee support direct cost, (6) losses of revenue due to the patients’ types seen in sites of teaching, (5) patients’ loss, and (7) clinical costs elevation due to patterns of students practice and the use of extra medical resources [17].

While there is existing literature on the different barriers to effective learning in ambulatory care clinics, many simply describe them and divide these issues into broader categories. This study aims to identify these barriers using qualitative analysis to investigate which specific factors were the most significant. The study was conducted in AU and KSU, two of the largest medical colleges in Saudi Arabia providing us with an adequate number of participants. By surveying both medical students and faculty members in these two major academic centers, we ensured factors affecting all parties involved in the explored learning process were mostly addressed.

### 4.1. Environmental Factors

The most significant environmental barrier to effective ambulatory clinic learning mentioned by both students and faculty members were restrictions put in place due to COVID-19. While this may seem to be a temporary barrier due to the current pandemic, it serves as a reminder of the considerable burden the pandemic or other infectious disease outbreaks has had on the medical education system [18,19,20]. It has been challenging for medical colleges to adapt to this unprecedented situation, with many opting for virtual learning. Harries et al. found that students recognized significant disruption to their medical education, with many willing to accept the risk of acquiring the infection and return to previous guidelines [21]. While the safety of both students and patients should be of utmost importance, innovative solutions must be developed to help minimize the disruption in the medical education system.

Medical students and faculty members also highlighted the need of optimizing the distribution of students and the importance of structured teaching objectives in clinics [14,22,23]. While these can be significant barriers, adjustments to both teaching curriculums and student schedules can ensure these barriers are reduced, providing a more structured and productive clinical teaching sessions to medical students. Studies suggest that allowing students the opportunity to talk to patients and perform clinical examinations alone as part of the learning objectives was greatly beneficial. Students reported increased learning opportunities as well as more confidence in clinical skills such as history taking and clinical examination [24,25].

### 4.2. Faculty Related Factors

Under faculty-related factors, many students and faculty members stated that the relatively insufficient time for teaching due to intense patients’ agenda was an important barrier to effective outpatient clinic teaching. Inadequate supervision by faculty members was another significant barrier perceived by students, which can be explained by the time constraints during most clinic visits. This is consistent with several studies on the topic which suggest that the time pressures of ambulatory clinics often hindered learning, especially in health institutions with high patient demands [26,27,28,29]. With their own clinical workload increasing and continued pressures to increase clinical productivity, balancing both patient care and student education can be extremely challenging for faculty members. Despite this, the literature report that most faculty members receive very little to no explicit training on the processes of teaching. Faculty development in the form of workshops and conferences could serve as an essential tool to aid in dealing with the complexities of medical teaching today [30]. In addition, a study looking into protecting teaching time in an outpatient clinic recommended that academic medical centers must reaffirm education as one of their central missions and ensure all policies are in line with this vital goal [31]. Emphasis must be placed on the importance of spending time on teaching during clinic visits to provide high-quality medical education to young physicians.

Some faculty members also mentioned the fact that teaching was not integrated as part of their KPI, and annual evaluation was another important barrier to effective teaching. This was an interesting finding as there are little to no studies that have cited the lack of ACC teaching on KPI as a significant barrier to effective teaching. However, this presents another opportunity for medical schools to clearly communicate and highlight the importance of ambulatory clinic teaching to their faculty members, in addition to finding a suitable way to minimize the negative impact of time spent on teaching on KPI. More research must be done to investigate whether adding key performance indicators related to teaching would improve outpatient education.

### 4.3. Patient Related Factors

Most students and faculty members agreed that patient refusal to be seen by medical students was an important barrier to effective ambulatory clinic education. However, this opposes other studies which suggest that only a minority of patients refuse or have negative feelings towards the involvement of medical students in their care [32,33,34,35]. Marwan et al. investigated factors that affect patient refusal and found that only a minority of participants would refuse students to take their medical history with the presence of a supervising doctor; however, the refusal rate was higher when the patients were asked if they would permit medical students to take their history unsupervised [32]. Faculty members and students should learn to master essential communication skills to aid in convincing patients and ensure that their refusal is not a barrier to learning.

Fear of patient dissatisfaction was another significant barrier that faculty members constantly faced. Patient satisfaction has become one of the cornerstones of modern medicine and has always been a top priority for all physicians and healthcare providers. Concerns for patient dissatisfaction have contributed to the unwillingness of many institutions and faculty members to embrace medical education [36]. Despite popular belief, studies looking into the effects of medical student teaching in an outpatient setting found that patient satisfaction was not affected by the presence of the students [36,37]. This further confirms that medical student education in ambulatory clinics should always be encouraged, and that patient refusal or fear of dissatisfaction should not be considered a barrier to effective learning.

Continuity of care is a fundamental component of primary healthcare and is associated with improved clinical outcomes and adherence to treatment regimens [38]. Despite this, maintaining adequate continuity for students to experience in ambulatory clinics remains a challenge, most notably, due to students’ division of time between inpatient wards, outpatient experiences, and other specialties [39]. We found that students and faculty members agreed that lack of continuity heavily affected learning in an ambulatory clinic setting which is consistent with the current literature [38,39,40].

Another significant barrier that students faced in clinics was the lack of suitable patients for teaching. This was an expected result since our study was centered around teaching in tertiary care hospitals, where most patients tend to have subspecialty conditions that may be less suitable for undergraduate teaching. Several studies also stressed the importance of selecting suitable patients in accordance with the student’s level and learning objectives to maximize effective learning [15,22].

### 4.4. Student Related Factors

We found that an increasing number of students attending clinics was an important barrier that significantly compromised ambulatory clinic education which is consistent with many studies [3,41,42]. A higher number of students in a given clinic will affect the time and exposure for each individual student to spend with patients, an essential component of outpatient learning. This will also compromise the ability of faculty members to provide precise feedback and evaluations, which are an important part of effective learning. Logistical solutions must be put into place to ensure an adequate number of students are distributed to each clinic to facilitate a positive learning environment for both faculty members and students.

Lack of student’s commitment and interest in learning was a significant barrier that was mentioned by our faculty members. This was also noted in another study which found that students who did not take charge of their learning tend to show little interest in learning [41]. Azher et al. supported a style of outpatient teaching where students are not just observers but are actively involved in patient care, which was the preferred participation style from a student’s perspective and significantly improved their interest [43].

### 4.5. Study Implications

Learning in ambulatory care clinics has become an essential component of medical education that provides many unique learning opportunities. Compared to inpatient wards, ambulatory clinics provide students with an opportunity to encounter a plethora of typical conditions and presentations and provide students with vital training in communication skills, preventive medicine, and psychosocial elements of disease [27].

Despite these benefits, ambulatory clinics host several barriers which can greatly hinder the overall learning experience. While there are several studies that have described these barriers, few have taken an analytic approach to identify which factors have the single biggest impact on learning from both the tutor and the student’s perspective. Our study in two of the largest tertiary care hospitals in Saudi Arabia aimed to fill this gap and identified some of the biggest challenges faced when learning in ambulatory care clinics. Insufficient time, patient refusal, increased number of students, and restrictions during the COVID-19 pandemic were the biggest barriers we observed during our analysis. As the field of medicine is constantly evolving, the content and delivery of medical education continue to be globally refined. With these significant barriers identified, our findings can serve as a guide to medical educators and curriculum developers to ensure the necessary changes are made, to facilitate effective clinical teaching and the overall improvement of the medical education system.

### 4.6. Limitations

The two contributing teaching hospitals have a wide range of world-class facilities and infrastructure tailored for complex cases requiring tertiary care. This could serve as a point of bias, as most outpatient teaching is conducted in less specialized centers. Surveying students and faculty members from primary and secondary health care facilities would provide more generalizable results. The convenience sampling could have biased the collected samples, which may not necessarily be representative of the other students or faculty from either university. Moreover, a larger study may reveal other significant barriers which may not be present in these advanced tertiary care centers. Additionally, future studies should aim to include a larger sample size. Another limitation is that our study was carried out during the COVID-19 crisis, and quantitative data were not compiled from the experts’ feedback regarding the used instrument; therefore, the Q-sorting or Kendall’s W analysis was not used because much of the correspondence was lost during the pandemic crisis, so future research could explore if the reported barriers change after the pandemic is over.

## 5. Conclusions

Our cross-sectional study identified ambulatory clinic teaching barriers as reported by medical students and faculty members in two tertiary care hospitals. Insufficient time for teaching, patient refusal, and restrictions due to COVID-19 were the most significant barriers. Faculty members also highlighted that teaching was not integrated as part of the KPI. Identifying these factors paves the way for future studies to investigate these specific issues and help develop innovative solutions to eliminate these barriers.

While it is difficult to be generalized, the outcomes of this study give an idea about possible challenges that medical students might face when attending ACC. This study paves the way for future work, which can be conducted to evaluate these challenges on a bigger scale and in different hospitals all over the globe. Moreover, and based on our results, we think it is important for medical schools to understand all the challenges that are faced by their students through a validated and concise tool. This can highlight the major challenges and allow decision-makers to target them individually; aiming to improve medical education in ACC settings.

## Figures and Tables

**Figure 1 healthcare-10-00496-f001:**
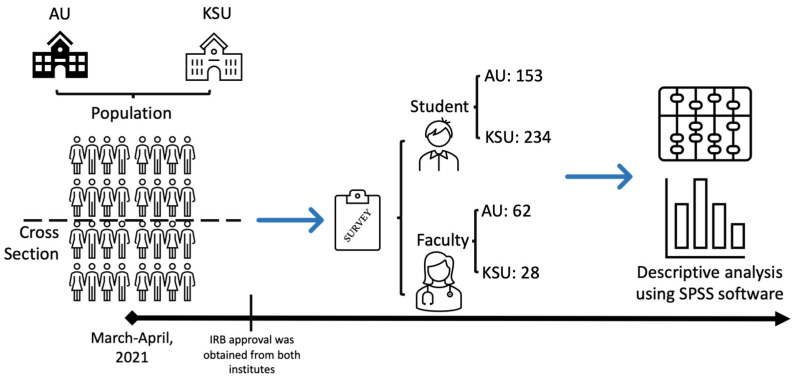
Flow diagram depicting the methodology we followed in this study.

**Figure 2 healthcare-10-00496-f002:**
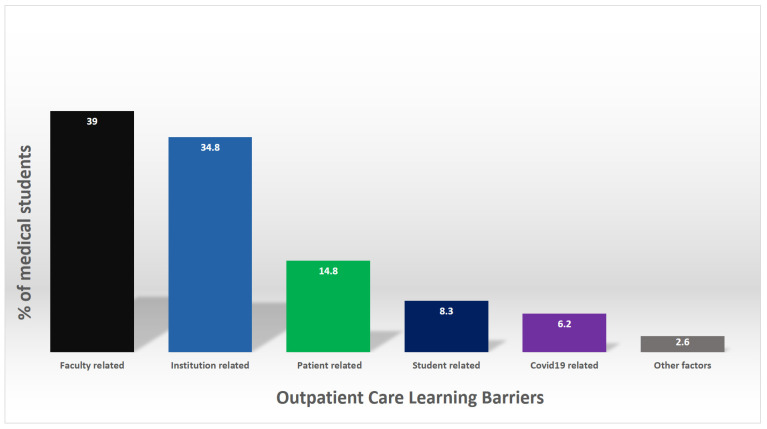
Medical students perceived barriers to ACC learning.

**Table 1 healthcare-10-00496-t001:** Descriptive analysis of medical students’ demographic (*n* = 387).

	Frequency	Percentage
**Sex**		
Female	177	45.7
Male	210	54.3
**Study year**		
4th	190	49.1
5th	197	50.9
**University**		
Alfaisal University	153	39.5
King Saud University	234	60.5
**With whom do you prefer attending clinics?**		
Consultant	171	44.2
Fellow	60	15.5
Junior Resident	58	15
Senior Resident	98	25.3
**What are the challenges to have proper and efficient outpatient education?**		
Faculty related	150	39
Institution related	134	34.8
Patient related	57	14.8
Student related	32	8.3
COVID-19 related	24	6.2
Other factors	10	2.6

**Table 2 healthcare-10-00496-t002:** Descriptive analysis of the medical students’ perceptions about virtual clinics and preferred clinics in ACC.

	Frequency	Percentage (%)
**Do you support involving medical students in virtual clinics?**		
To No extent	78	20.2
To little extent	59	15.2
To Moderate extent	98	25.3
To Great extent	59	15.2
To Very great extent	93	24
**What are the best clinics you have attended during your medical school?**		
Internal medicine and IM subspecialties	145	40.4
Oby-Gyn	75	20.9
Pediatrics	72	20.1
General surgery	68	18.9
Ophthalmology	53	14.8
Orthopedics	50	13.9
Family medicine	41	11.4
ENT	32	8.9
Psychiatry	25	7
Neuroscience	9	2.5
Other clinics	6	1.7
Vascular and thoracic surgery	1	0.3

**Table 3 healthcare-10-00496-t003:** Factors considered by medical students as barriers for ACC education.

	Frequency (%)
**Environment-related factors**	
- Restrictions due to COVID-19	203 (52.5)
- Lack of structured teaching objectives	94 (24.3)
- Inadequate distribution of student in ambulatory clinics	74 (19.1)
- Inappropriate or small clinics rooms	16 (4.1)
**Faculty-related factors**	
- Inadequate supervision and teaching by faculty	188 (48.6)
- Lack of time for teaching by doctor due to intense patient agenda	131 (33.9)
- Inappropriate or absence of feedback	47 (12.1)
- Fear of losing private patients	21 (5.4)
**Patient-related factors**	
- Patients’ refusal to be seen by medical students	192 (49.6)
- No follow-up/continuity of care for cases attended	98 (25.3)
- Lack of suitable patients for teaching	97 (25.1)
**Student-related factors**	
- The increasing number of students attending ambulatory clinics	151 (39)
- Not enough time to attend the clinics	118 (30.5)
- Lack of student’s commitment and interest in learning	72 (18.6)
- No additional information gained compared to inpatient setting	46 (11.9)

**Table 4 healthcare-10-00496-t004:** Bivariate comparison of the students from the two universities on their perceived learning preferences and barriers.

	AU *n* = 153	KSU, *n* = 234	Test Statistic	*p*-Value
**Sex**				
Female	95 (62.1)	82 (35)	χ^2^ (1) = 27.3	<0.001
Male	58 (37.9)	152 (65)		
**Study year**				
4th	83 (54.2)	107 (45.7)	χ^2^ (1) = 2.67	0.101
5th	70 (45.8)	127 (54.3)		
**As a medical student, with whom do you prefer attending clinics?**				
Consultant	60 (39.2)	111 (47.4)	χ^2^ (3) = 7.50	0.057
Fellow	33 (21.6)	27 (11.5)		
Junior resident	22 (14.4)	36 (15.4)		
Senior resident	38 (24.8)	60 (25.6)		
**During your training in 4th/5th year, what are the barriers and challenges for medical students to have proper and efficient outpatient education?**				
Faculty related	70 (45.8)	80 (34.2)	χ^2^ (1) = 5.22	0.022
Patient related	26 (17)	31 (13.2)	χ^2^ (1) = 1.04	0.309
Student related	13 (8.5)	19 (8.1)	χ^2^ (1) = 0.017	0.895
Institution related	35 (22.9)	99 (42.3)	χ^2^ (1) = 15.43	<0.001
COVID-19 related	4 (2.6)	20 (8.5)	χ^2^ (1) = 5.60	0.018
Other factors	4 (2.6)	6 (2.6)	χ^2^ (1) = 0.002	0.976
**Do you support involving medical students in virtual clinics? -Mean (SD)**	3.28 (1.49)	2.94 (1.40)	t(309.7) = 2.23	0.026
**What do you think are the best clinics you have attended during your medical school?**				
Internal Medicine and IM subspecialties	76 (49.7)	69 (29.5)	χ^2^ (1) = 16.1	<0.001
Oby-Gyn	40 (26.1)	35 (15)	χ^2^ (1) = 7.41	0.006
Orthopedics	3 (2)	47 (20.1)	χ^2^ (1) = 27.01	0.001
Psychiatry	1 (0.7)	24 (10.3)	χ^2^ (1) = 14.1	<0.001
Pediatrics	58 (37.9)	14 (6)	χ^2^ (1) = 62.3	<0.001
**ENVIRONMENTAL-RELATED FACTORS**				
Inadequate distribution of student in ambulatory clinics	23 (15)	51 (21.8)	χ^2^ (3) = 25.8	<0.001
Inappropriate or small clinics rooms	6 (3.9)	10 (4.3)		
Lack of structured teaching objectives	58 (37.9)	36 (15.4)		
Restrictions due to COVID-19	66 (37.9)	137 (58.5)		
**FACULTY-RELATED FACTORS**				
Fear of losing private patients	13 (8.5)	8 (3.4)	χ^2^ (3) = 4.91	0.179
Inadequate supervision and teaching by faculty	73 (47.7)	115 (49.1)		
Inappropriate or absence of feedback	19 (12.4)	28 (12)		
Lack of time for teaching by doctor due to intense patient agenda	48 (31.4)	83 (35.5)		
**PATIENT-RELATED FACTORS**				
Lack of suitable patients for teaching	35 (22.9)	62 (26.5)	χ^2^ (2) = 3.10	0.216
No follow-up/continuity of care for cases attended	46 (30.1)	52 (22.2)		
Patients’ refusal to be seen by medical students	72 (47.1)	120 (51.3)		
**STUDENT-RELATED FACTORS**				
Increasing number of students attending ambulatory clinics	41 (26.8)	110 (47)	χ^2^ (3) = 21.81	<0.001
Lack of student’s commitment and interest in learning	36 (23.5)	36 (15.4)		
No additional information gained compared to inpatient setting	28 (18.3)	18 (7.7)		
Not enough time to attend the clinics	48 (31.4)	70 (29.9)		

**Table 5 healthcare-10-00496-t005:** Descriptive analysis of the faculty’s demographic and professional characteristics.

	Frequency (90)	Percentage (%)
**S** **ex**		
Female	15	16.7
Male	75	83.3
**Age group**		
30–39 years	15	16.7
40–49 years	32	35.6
50–59 years	32	35.6
60 years and above	11	12.2
**University**		
Alfaisal University	62	68.9
King Saud University	28	31.1
**Academic title**		
Assistant professor	30	33.3
Associate professor	32	35.6
Lecturer	4	4.4
Professor	24	26.7
**Specialty**		
Internal medicine	39	43.3
Pediatrics	21	23.3
Neurosciences	6	6.7
Emergency medicine	5	5.6
Surgical subspecialties	4	4.4
ENT	3	3.3
Oby-Gyn	3	3.3
Family medicine	2	2.2
Ophthalmology	2	2.2
Psychiatry	2	2.2
Oncology	1	1.1
Orthopedics	1	1.1
Radiology	1	1.1

**Table 6 healthcare-10-00496-t006:** Descriptive analysis of the general perceptions of faculty towards the barriers in medical students teaching.

Frequency	Percentage (%)
How comfortable are you if a medical students attend ACC with you?	
Uncomfortable	35.6
Somehow comfortable	25.6
Moderately	11.1
Comfortable	16.7
Very comfortable	11.1
Outpatient experience is important for medical students and it should be integrated in the curriculum.	93.3
What are the main challenges & barrier to have proper and efficient medical student education in ACC settings?	
Environment/institutional-related	55.6
Patient-related	24.4
Faculty-related	20
Student-related	16.7
Time-related	6.7
How much do you support involving medical student in virtual clinics?	
To No extent	20
To little extent	6.7
Some extent	18.9
Great extent	17.8
Very great extent	36.7

**Table 7 healthcare-10-00496-t007:** The faculty perceived order (priority) of medical teaching barriers and challenges in outpatient settings.

	Frequency (%)
**Environment-related factors**	
Restrictions due to COVID-19	(26.7)
Inadequate structuring and distribution of student in ACC	(21.1)
Inappropriate or small clinics rooms	(21.1)
Lack of institutional support	(18.9)
Inadequate financial incentives for academic staff	(12.2)
**Faculty-related factors**	
Insufficient time for teaching due to intense patient agenda	(70)
Not integrated as part of my current KPI	(20)
Not feeling comfortable to have students in my clinic	(7.8)
Lack of training for faculty to teach medical students in ACC	(2.2)
**Patient-related factors**	
Patients’ refusal to be seen by medical students	(41.1)
Fear of patient dissatisfaction	(24.4)
No follow-up/continuity of care for cases attended	(21.1)
Lack of suitable patients for teaching	(13.3)
**Student-related factors**	
Increasing numbers of students	(48.9)
Lack of student’s commitment and interest in learning	(51.1)

**Table 8 healthcare-10-00496-t008:** Bivariate comparison of the teacher from the two universities on their perceived learning preferences and barriers.

	University			
	AU (*n* = 62)	KSU (*n* = 28)	Test Statistic	*p*-Value
**Sex**				
Female	12 (19.4)	3 (10.7)	χ^2^ (1) = 0.51	0.476
Male	50 (80.6)	25 (89.3)		
**Age group**				
30–39 years	10 (161)	5 (17.9)	χ^2^ (3) = 4.60	0.204
40–49 years	18 (29)	14 (50)		
50–59 years	25 (40.3)	7 (25)		
60 years and above	9 (14.5)	2 (7.1)		
**Academic title**				
Assistant professor	26 (41.9)	4 (14.3)	χ^2^ (3) = 13.12	0.004
Associate Professor	16 (25.8)	16 (57.1)		
Lecturer	4 (6.5)	0		
Professor	16 (25.8)	8 (28.6)		
**How comfortable you are if a medical students attend ACC with you? mean (SD)**	2.40 (1.49)	2.46 (1.2)	t(64.23) = 0.21	0.837
**Do you think outpatient experience is important for medical students** **and it should be integrated in the curriculum?**				
No	2 (3.2)	4 (14.3)	χ^2^ (1) = 2.22	0.136
Yes	60 (96.8)	24 (85.7)		
**From your experience, what are the main challenges/barrier to have proper and efficient medical student education in ACC settings?**				
Environment/institutional-related	32 (51.6)	18 (64.3)	χ^2^ (1) = 1.26	0.263
Faculty-related	10 (16.1)	8 (28.6)	χ^2^ (1) = 1.87	0.172
Patient-related	14 (22.6)	8 (28.6)	χ^2^ (1) = 0.376	0.541
Student-related	10 (16.1)	5 (17.9)	χ^2^ (1) = 0.041	0.839
Time-related	4 (6.5)	2 (7.1)	χ^2^ (1) < 0.001	1.000
**How much do you support involving medical student in virtual clinics? mean (SD)**	3.66 (1.48)	2.96 (1.55)	t(88) = 2.04	0.045
**ENVIRONMENTAL-RELATED FACTORS**				
Inadequate financial incentives for academic staff	10 (16.1)	1 (3.6)	χ^2^ (4) = 5.65	0.227
Inadequate structuring and distribution of student in OPD	15 (24.2)	4 (14.3)		
Inappropriate or small clinics rooms	11 (17.7)	8 (28.8)		
Lack of institutional support	11 (17.7)	6 (21.4)		
Restrictions due to COVID-19	15 (24.2)	9 (32.1)		
**FACULTY-RELATED FACTORS**				
Insufficient time for teaching due to intense patient agenda	43 (69.4)	20 (71.4)	χ^2^ (3) = 9.15	
Lack of training/retraining for faculty to teach medical students in ACC	1 (1.6)	1 (3.6)		
Not feeling comfortable to have students in my clinic	2 (3.2)	5 (17.6)		
Not integrated as part of my current KPI	16 (25.8)	2 (7.1)		
**PATIENT**-**RELATED FACTORS**				
Fear of patient dissatisfaction	16 (25.8)	6 (21.4)	χ2 (3) = 1.34	0.72
Lack of suitable patients for teaching	9 (14.5)	3 (10.7)		
No follow-up/continuity of care for cases attended	14 (22.6)	5 (17.9)		
Patients’ refusal to be seen by medical students	23 (37.1)	14 (50)		
**STUDENT-RELATED FACTORS**				
Increasing numbers of students	31 (50)	13 (46.4)	χ2 (1) = 0.10	0.754
Lack of student’s commitment and interest in learning	31 (50)	15 (53.6)		

## Data Availability

Data available on request.

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
