# Peer review of "Challenges Facing Undergraduate Medical Education in Ambulatory Care Clinics at Tertiary Care Hospitals"

_healthcare, 2022, doi:10.3390/healthcare10030496_

Round 1

Reviewer 1 Report

Medical education has been rapidly growing and transforming due to the enormous evolution of medicine. There have been many proficient ways to learn in medicine, but  academic lectures, attending wards, and Ambulatory Care Clinics (ACC) remain the three main  ways of gaining clinical knowledge and experience for medical students. In the past decade, there  was a dramatic shift in care by focusing on ambulatory care rather than inpatient settings, which  provides a golden opportunity to reinforce medical education.

The most of the published studies that have focused on the teaching barriers in ACC were descriptive rather than analytic studies.

The  aim of the authors was to detect and determine the barriers to teaching in ACC settings using qualitative analysis.

The authors conclude that:

  • undergraduate medical education in outpatient settings has many challenges.
  • In their study, the most significant challenges were COVID-19 related restrictions, patient refusal, and insufficient time for teaching.
  • Future studies are needed to investigate these barriers and explore potential solutions that can decrease their burden on undergraduate medical education.

The study has merits, however it needs improvements.

Please answer to the following suggestions / comments:

1) Please illustrate the purpose in more detail, possibly separating it from the introduction

2) The methods are listed as a list. They need to be better harmonized and possibly supported by a flow-chart.

3) Some ethical aspects must be included in the appropriate declaration after the conclusions

4) Even the results should be improved like the methods, by inserting, for example, before the thematic analysis a summary of what will be said.

5) The conclusions should be expanded with perspectives / future work

Author Response

MDPI Office

Ref.: Manuscript ID Healthcare-1607805                                                                 

Dear MDPI Office,

Thank you for giving us the opportunity to revise our manuscript. We also would like to thank the editorial board and the reviewers for their valuable feedback. We reviewed the manuscript and the suggestions carefully and made changes accordingly. The suggestions given have definitely contributed to improving the manuscript.

We just want to highlight that we mistakenly submitted our study as “Study Protocol” instead of “Original Article”. We highlighted this in the edited version.

Here is the point-by-point response to all the comments.

Reviewers’ Comments:

Reviewer #1

Please illustrate the purpose in more detail, possibly separating it from the introduction.

Response: Great suggestion. We followed the reviewer’s suggestion and separated the purpose of our study. We also elaborated more on our aims. The aims can be found, as a separate paragraph, in the last part of the introduction.

The methods are listed as a list. They need to be better harmonized and possibly supported by a flow-chart.

Response: As per the reviewer suggestion, we added Figure 1 as a flow diagram to show the methodology we followed.

Some ethical aspects must be included in the appropriate declaration after the conclusions.

Response: Thank you for the suggestion. The study was approved by IRB at both institutions and participation in the study was voluntary and anonymous. In addition, consent to participate is incorporated in the survey cover page, and data was stored under password protection and this was mentioned in “ethical considerations”. We appreciate further elaboration, if the reviewer would like us to add or edit.

Even the results should be improved like the methods, by inserting, for example, before the thematic analysis a summary of what will be said.

Response: Thank you for the suggestion. We added a short paragraph at the beginning of the results section to briefly mention what we are going to discuss. We also made sure that both sections 3.1 and 3.2 have similar arrangement (except for one extra section, 3.1.3, which can only be found in student section, 3.1).

Even the results should be improved like the methods, by inserting, for example, before the thematic analysis a summary of what will be said.

Response: We totally agree. We added a new paragraph in the conclusion section to suggest two things, future work and how these results could be utilized to possibly improve medical education at ACC.

Again, we thank everyone who took time to review this manuscript.

We are re-submitting the manuscript as per your instructions.

Sincerely,

Youssef B. Almushait

Mohamad S. Alabdaljabar

Khalid Alkhani

Hesham M. Abdalla

Raid Alhayaza

Hani Temsah

Fahad Alsohaibani

Reviewer 2 Report

This study, developed in the field of medical education, seeks to describe the teaching barriers identified by senior medical students and faculty. This cross-sectional study, based on the analytical categories defined by Franco et al., included a representative sample from two university institutions.

Despite the interest of the study, the following corrections and improvements are necessary:

(a) Sample. Describe the sociodemographic characteristics in subsection 2.1. and merge it with subsection 2.2. Some of this information is erroneously included in section 3.

b) Replace the label "Data collection" with "Instrument". In this subsection, provide the empirical evidence corresponding to its validation.

c) Incorporate a sub-section focused on the procedure followed.

d) In subsection 2.4., indicate that a regression analysis has been applied. However, its purposes are not specified and its results are not provided. 

Author Response

18th Feb 2022

MDPI Office

Ref.: Manuscript ID Healthcare-1607805                                                                 

Dear MDPI Office,

Thank you for giving us the opportunity to revise our manuscript. We also would like to thank the editorial board and the reviewers for their valuable feedback. We reviewed the manuscript and the suggestions carefully and made changes accordingly. The suggestions given have definitely contributed to improving the manuscript.

We just want to highlight that we mistakenly submitted our study as “Study Protocol” instead of “Original Article”. We highlighted this in the edited version.

Here is the point-by-point response to all the comments.

Reviewers’ Comments:

Reviewer #2                                                                 

Sample. Describe the sociodemographic characteristics in subsection 2.1. and merge it with subsection 2.2. Some of this information is erroneously included in section 3.

Response: Great point. While both approaches can be taken, we think that including the results of the survey, including the sociodemographic characteristics of responders, is better suited for section 3 (Results), rather than section 2 (Methodology).

Replace the label "Data collection" with "Instrument". In this subsection, provide the empirical evidence corresponding to its validation.

Response: We edited the title of the subsection as suggested. As for tool validation, we could not find a tool to specifically assess challenges in ACC for medical students. Thus, we created a tool and did a preliminary analysis of the initial responses that we got. We later excluded these responses before distributing the questionnaire on a wide range. We elaborated on this in the methods section.

Incorporate a sub-section focused on the procedure followed.

Response: Based on the suggestion for our respectful reviewers, we added Figure 1 as a flow diagram to show the method that we followed. We would be happy to make further modifications, if needed.  

In subsection 2.4., indicate that a regression analysis has been applied. However, its purposes are not specified and its results are not provided.

Response: Thank you for pointing this out. Regression analysis was removed as it is not applicable to our data.

Again, we thank everyone who took time to review this manuscript.

We are re-submitting the manuscript as per your instructions.

Sincerely,

Youssef B. Almushait

Mohamad S. Alabdaljabar

Khalid Alkhani

Hesham M. Abdalla

Raid Alhayaza

Hani Temsah

Fahad Alsohaibani

Round 2

Reviewer 1 Report

The manuscripr has strongly improved.

I have not further comments

Author Response

MDPI Office

Ref.: Manuscript ID Healthcare-1607805                                                                 

Dear MDPI Office,

Thank you for giving us the opportunity to revise our manuscript. We also would like to thank the editorial board and the reviewers for their valuable feedback. We reviewed the manuscript and the suggestions carefully and made changes accordingly. The suggestions given have definitely contributed to improving the manuscript.

We just want to highlight that we mistakenly submitted our study as “Study Protocol” instead of “Original Article”. We highlighted this in the edited version.

Please note that text in red represents the edits that we did in revision, round 1, whereas text highlighted in yellow represents the changes we did in round 2.

Here is the point-by-point response to all the comments.

Reviewers’ Comments:

Reviewer #1

The manuscript has strongly improved. I have not further comments.

Response: Thank you very much for your time and valuable input.

Reviewer #2                                                                 

Thank you very much for attending to my suggestions, aimed at improving the impact and methodological rigor of your research.

Response: Thank you our dear reviewer for your time and valuable suggestions.

Since the instrument, designed ad hoc, is measured on a Likert scale and an inter-judge agreement evaluation has been performed, it would be necessary to include empirical evidence on the validation process in the specific context of this research. In the first case, at least through the Cronbach's a calculation and, in the second, through inter-expert agreement indexes.

Response: Thank you for pointing this out. However, the questionnaire was not measured with a Likert-Like metric scale/questionnaire, it was measured with multiple-response dichotomous questions (tick all that applies to you) of barriers, as follows: applies to me/does not apply to m). The Cronbach's alpha applies only to Likert-Like scales of 1-4 levels, it does Not suite binary questions. Even if we apply the Kuder-Richardson's test KR-20 test it would end up in misuse of the statistics, unless you believe that the face and content validity, we have conducted on the questionnaire was not sufficient, too many quantitative and qualitative questionnaires shed useful light on various phenomena even if they are not measured with Likert-Like scales. There is one question only (i.e., virtual clinic questions) indeed that was measured with Likert-like scale in both surveys, and it cannot be tested for reliability because it is a single item. Thank you again for your comment, and if you believe the KR-20 test suites the situation we do not really have a problem conducting it, but generally it isn't suitable/applicable for similar measured aspects.

Finally, it would be very useful to include the complete applied instrument as an appendix to this study, so that it can be replicated.

Response: Great suggestion. We have now included both surveys in the supplementary material as suggested.

Again, we thank everyone who took time to review this manuscript.

We are re-submitting the manuscript as per your instructions.

Sincerely,

Youssef B. Almushait

Mohamad S. Alabdaljabar

Khalid Alkhani

Hesham M. Abdalla

Raid Alhayaza

Mohamad-Hani Temsah

Fahad Alsohaibani

Reviewer 2 Report

Dear Authors,

Thank you very much for attending to my suggestions, aimed at improving the impact and methodological rigor of your research.

Although the study has improved significantly, please appreciate and incorporate the following concluding remarks:

Since the instrument, designed ad hoc, is measured on a Likert scale and an inter-judge agreement evaluation has been performed, it would be necessary to include empirical evidence on the validation process in the specific context of this research. In the first case, at least through the Cronbach's a calculation and, in the second, through inter-expert agreement indexes.

Finally, it would be very useful to include the complete applied instrument as an appendix to this study, so that it can be replicated.

We look forward to this new revision.

Best regards,

Reviewer 2

Author Response

(The authors gave the same response as above.)

Round 3

Reviewer 2 Report

Dear Authors,

Thank you for responding to my concerns, which have been amply clarified in your explanations and in the supplementary files.

Indeed, my doubts came from the question related to virtual clinic sessions. This is a single item (section 3.1.3), so it is not necessary to calculate its reliability and internal consistency.  In this case, it would only be appropriate to provide the necessary evidence on compliance with parametric assumptions before applying Student's t test.

Although the scientific literature provides solid theoretical bases for the construction of the instrument applied, it would have been desirable to provide empirical evidence on the validity of the questionnaire. Given the mostly categorical nature of its construction, content validity by means of an inter-judge analysis (degrees of agreement) would have been desirable (e.g., Kendall's W, among many others, always depending on the measure of the rating scale applied). However, this question does not reduce the overall quality of the study submitted.

Best regards,

Reviewer 2

Author Response

Response: Dear Reviewer 2,

Thank you for your valuable comments.

We have added that “five out of the six experts who evaluated the content and the face validity of the instrument had agreed on its value and usefulness beside the relevance of each barriers and facilitators we have measured. Their feedback was measured independently via electronic documents and emails.”

We were adjusting the items and wording of the tool according to their sent feedback. Unfortunately, we did not compile quantitative data on their feedback, therefore, the Q-sorting and Kendall’s Tau analysis are not available for us at the time being because many of their correspondence were lost during the hectic pandemic crisis. We value your informative feedback, that’s why we have added this in our study limitation section as we could not do such analysis. Also, For the t-test, could you kindly note us where the statistical assumptions for the t-test in the analysis findings were violated? For further clarification, the t-test shows two outputs for equality of variance and violated equality of variance, and to our best knowledge, we have quoted the right analysis findings for the given situation.

We thank you for your time and your valuable feedback.

Again, we thank everyone who took time to review this manuscript.

We are re-submitting the manuscript as per your instructions.

Sincerely,

Youssef B. Almushait

Mohamad S. Alabdaljabar

Khalid Alkhani

Hesham M. Abdalla

Raid Alhayaza

Mohamad-Hani Temsah

Fahad Alsohaibani

Round 4

Reviewer 2 Report

Dear authors,

Thank you very much for heeding my recommendations, which are aimed at increasing the value and scientific impact of your article.

The inclusion of the statistical analysis of inter-judge agreement (e.g., Kendall's W -not Kendall's tau b-) in the study limitations is appropriate and correct.

As a point of clarification, the application of parametric statistics (Student's t test) requires the demonstration of normality in the distribution of the data (dependent variable), together with homoscedasticity between groups. However, this test, as very well indicated, offers an alternative in case of non-compliance. Although I consider that this point is not fundamental to increase the value of this research as a whole, could you please state it briefly in your manuscript (normal distribution of the data in the dependent variable). Thus, the reader will not wonder about this requirement.

Best regards and congratulations,

Reviewer 2

Author Response

3 March 2022

MDPI Office

Ref.: Manuscript ID Healthcare-1607805                                                                 

Dear MDPI Office,

Thank you for giving us the opportunity to revise our manuscript. We also would like to thank the editorial board and the reviewers for their valuable feedback. We reviewed the manuscript and the suggestions carefully and made changes accordingly. The suggestions given have definitely contributed to improving the manuscript.

We just want to highlight that we mistakenly submitted our study as “Study Protocol” instead of “Original Article”. We highlighted this in the edited version.

Here is the point-by-point response to all the comments.

Reviewer #2

Dear authors,

Thank you very much for heeding my recommendations, which are aimed at increasing the value and scientific impact of your article.

The inclusion of the statistical analysis of inter-judge agreement (e.g., Kendall's W -not Kendall's tau b-) in the study limitations is appropriate and correct.

As a point of clarification, the application of parametric statistics (Student's t test) requires the demonstration of normality in the distribution of the data (dependent variable), together with homoscedasticity between groups. However, this test, as very well indicated, offers an alternative in case of non-compliance. Although I consider that this point is not fundamental to increase the value of this research as a whole, could you please state it briefly in your manuscript (normal distribution of the data in the dependent variable). Thus, the reader will not wonder about this requirement.

Best regards and congratulations,

Reviewer 2

Response: Dear Reviewer 2,

Thank you for your valuable comments that have significantly improved our paper.

We have added the kendall’s W in our limitation as suggested. Also, we have demonstrated the normal distribution of the data in the dependent variable, in the data analysis section 2.4 as suggested.

Again, we thank everyone who took time to review this manuscript.

We are re-submitting the manuscript as per your instructions.

Sincerely,

Youssef B. Almushait

Mohamad S. Alabdaljabar

Khalid Alkhani

Hesham M. Abdalla

Raid Alhayaza

Mohamad-Hani Temsah

Fahad Alsohaibani

This manuscript is a resubmission of an earlier submission. The following is a list of the peer review reports and author responses from that submission.